# The role of online peer support in pregnancy: A scoping review

Honor Morris[1], Maria Duaso[1], Marilyn Ijeomah-Orji[2], Lisa Akester[3], Jenny Roddy [4], Jayne Samples[5], Kate Heighway[6], Nigel Simpson[4,7], Tomasina Stacey [1]*

1 Faculty of Nursing Midwifery and Palliative Care, Kings College London, London, United Kingdom, 2 Imperial College Healthcare NHS Trust, London, United Kingdom, 3 Auntie Pam's, Kirklees Council, Kirklees, United Kingdom, 4 Leeds Teaching Hospitals Trust, Leeds, Leeds, United Kingdom, 5 University of Huddersfield, Huddersfield, United Kingdom, 6 Calderdale and Huddersfield Foundation Trust, Halifax, United Kingdom, 7 Department of Women's & Children's Health, Leeds Institute of Medical Research, University of Leeds, Leeds, United Kingdom

* Tomasina.stacey@kcl.ac.uk

## Abstract

### Background

Online peer support groups offer virtual spaces where members can share experiences, seek advice, and offer mutual support. While in-person peer support in pregnancy has shown benefits such as improved well-being and reduced isolation, limited research has explored online peer support groups. This scoping review aims to examine the types, contexts, and perceived benefits of online peer support for pregnant people.

### Methods

A systematic search was conducted in five databases (Embase, Medline PsychInfo, MIDIRS and CINAHL) from inception to December 2024. Eligible studies examined online peer support accessed by pregnant participants with direct interaction. Thematic analysis was applied to identify structural factors influencing the functioning of online peer support groups. The Langford Conceptual Analysis of Social Support was used to categorise type of support.

### Findings

A total of 27 studies were included. The findings highlighted several structural factors that influence the functioning of online peer support groups for pregnant people. Group size was identified as a key factor, with smaller groups promoting more meaningful interactions. Active moderators supported participant interaction and ensured information quality, while privacy settings, such as closed groups and anonymised participation, fostered safe discussions. Emotional support, such as expressions of empathy, reassurance, and encouragement, was the most reported benefit, followed

**Data availability statement:** All relevant data are within the manuscript and its Supporting Information files.

**Funding:** This work was supported by the National Institute of Health research: NIHR 205347 The funders had no role in data collection, analysis, interpretation, report writing, or the decision to submit this report for publication.

**Competing interests:** The authors have declared that no competing interests exist.

by informational and appraisal support. Instrumental support was observed less frequently.

## Conclusion

This scoping review identified key structural factors shaping the design and functioning of online peer support groups for pregnant people. Group size, active moderation, and privacy settings influenced participation and engagement. Further research is recommended to explore how these structural elements impact long-term engagement and overall effectiveness for pregnant people.

This project is funded by the National Institute for Health and Care Research (NIHR) under its Research for Patient Benefit (RfPB) Programme (Grant Reference Number NIHR205347). The views expressed are those of the author(s) and not necessarily those of the NIHR or the Department of Health and Social Care.

## Introduction

As virtual communication and interaction becomes more commonplace, there is a growing acknowledgement that online communities create a space for otherwise disparate individuals with common situations, such as pregnancy, to gather virtually and share experiences, seek information and provide emotional support or self-help [1,2]. Mutual relationship building and assistance within online communities can foster peer-to-peer support. This is a form of non-professional support engaged in by two or more individuals who understand one another's situation empathically through shared experience, enabling them to offer and receive help, comprising emotional, informational and practical elements [3]. It is founded on the key principles of respect, shared responsibility and a mutual understanding of what is beneficial to both parties [3].

Current evidence suggests that support provided by peers has proved beneficial for improving maternal quality of life and well-being. McLeish and Redshaw (2017) in their qualitative study of pregnant people receiving one-to-one peer support found that it '*contribute[d] to reducing their low mood and anxiety by overcoming feelings of isolation, disempowerment and stress, and increasing feelings of self-esteem, self-efficacy and parenting competence.*'[4], [p12]. Similarly, Renbarger et al (2021) found that group prenatal care '*facilitated the attainment of new knowledge and the formation of positive relationships with health care providers and peers.*' [5], [p159]. In-person support is widely used in healthcare and the National Institute for Health and Care Excellence (NICE) recommend that individuals should be advised of the potential benefits of peer support during pregnancy [6]. It is hypothesised that peer support (through peer-to-peer sharing) can help increase a sense of self efficacy and social connection [7]. Digital interventions, utilising social media, can facilitate the provision of peer-to-peer support in an accessible way, and can mitigate the complexity of setting up and providing face-to-face support. Although existing reviews explore the use of online peer support in other populations, none have specifically addressed its use during pregnancy. Therefore, this review aims to explore the types, contexts,

and perceived benefits of online peer support for pregnant people. The review will identify how these platforms are used, the forms of support exchanged, and key factors shaping their design and engagement.

## Methods

### Study design

This review was informed by the JBI methodology for scoping reviews [8] and follows the PRISMA-scoping (PRISMA-ScR) reporting guideline [9]. The completed PRISMA-ScR checklist is available in the supplementary file (S1 File). A protocol was registered on Open Science Framework Registries (https://doi.org/10.17605/OSF.IO/EKTZ3).

### Search strategy

A comprehensive search was initially conducted by one reviewer across five databases with no limitation on publication year: Embase (1974–22 April 2024), Ovid Medline (1946–22 April 2024), APA PsychInfo (1806 to April week 4 2024), Maternity & Infant Care Database (MIDIRS) (1971–22 April 2024), and CINAHL (1976 to April 2024). The search was updated by two further reviewers on 20th December 2024 prior to final analysis. Any conflicts were resolved by a third reviewer. This brings the total of individuals involved in the methodological assessment process to four. The search strategy and keywords were reviewed and refined in collaboration with two research librarians and three authors (HM, TS, MD) to ensure relevance and accuracy. The key concepts used in the search were variations on 'Online Social Network', 'Peer Support' and 'Pregnancy', please see S2 File for full search strategy.

### Study selection and data extraction

The inclusion and exclusion criteria (outlined in Table 1) followed the Population, Concept, and Context Framework and studies were included that met the following criteria: published studies that included pregnant people as the main population accessing online peer groups, and where participants in the groups were able to directly interact with each other, either openly or anonymously. Studies that focused on the development of an app or website for peer support or where the contents of the social media page were the main focus were not included.

Results were imported into Covidence (https://www.covidence.org). Duplicates were removed, after which titles and abstracts of the remaining papers were screened. All papers were screened independently by at least two authors and conflicts were resolved by mutual consent.

**Table 1. Inclusion and exclusion criteria.**

|  | Inclusion Criteria | Exclusion Criteria |
|---|---|---|
| **Participants** | Studies where the research population is pregnant people. | Studies where the research population is postnatal only. |
| **Concept** | Peer support | Studies where the main focus is to analyse the contents of a social media page or other online pages without examining the effect of online-based peer support. |
|  |  | Does not examine and/or analyse peer-to-peer support. |
| **Context** | Online platforms | Face to face support |
| **Type of study** | Primary research – all study designs were included as they could offer different insights to help us answer the research question | Reviews commentary, and other opinion pieces |
| **Language** | English language |  |

A standardised data extraction form, developed and tested by the research team within the Covidence data extraction tool (S3 File), was used to ensure consistency and accuracy. Two authors independently (MIO, MD or HM) extracted data, discrepancies were resolved through discussion. Key data items charted included study design, population characteristics, recruitment methods, virtual platform type, the presence of active moderators, and the types of support exchanged within the peer groups. To map data according to an established framework or theory as suggested by Pollock et al., [10] we used the Langford Conceptual Analysis of Social Support [11] which defines four domains: emotional support (expressions of empathy, care, and reassurance), informational support (sharing advice or knowledge), appraisal support (affirmation and constructive feedback), and instrumental support (tangible assistance or services). Using this framework allowed for the systematic categorisation and analysis of different types of social support that are exchanged in online peer support groups for pregnant people, providing a comprehensive understanding of how these groups function and their potential benefits.

### Critical appraisal assessment

We used critical appraisal tools to assess the reliability, validity, and applicability of the evidence. The CASP tool was used for randomised controlled trials (RCTs) and qualitative studies, the JBI Checklist for quasi-experimental designs, and the Mixed Methods Appraisal Tool (MMAT) [12]. For studies with cross-sectional observational designs, the quantitative section of the MMAT was applied. Including critical appraisal tools allowed us to highlight variations in study quality that could influence the interpretation of findings, providing a clearer understanding of the robustness of the evidence and enhancing the interpretation of the findings.

### Synthesis of results

We followed the JBI methodology and principles outlined by Pollock et al starting with frequency counts to quantify important features of the included studies [10]. Tabular synthesis was utilised to provide a comprehensive overview and highlight the key features of all included studies, facilitating easier comparisons and providing insight into the volume and type of existing research available.

The synthesis process involved a qualitative thematic analysis to identify key factors influencing the functioning of online peer support groups for pregnant people. An iterative approach was used to compare studies and categorise data, with themes emerging from patterns identified across the included studies. To classify the types of support exchanged, we applied Langford et al.'s conceptual framework (11), This process was guided by both numerical data and qualitative insights, allowing for a structured examination of the relationships between study characteristics, intervention components, and support mechanisms.

This mixed-methods synthesis, combining frequency counts, tabular presentation, and qualitative thematic analysis, provided a comprehensive overview of both structural factors and support dynamics. The systematic approach allowed for a balanced representation of both numerical patterns and contextual insights across the included studies

### Results

The electronic searches yielded a total of 2019 papers. There were 861 duplicates that were removed, leaving 1158 papers for the title and abstract screening. After title and abstract screening, 1045 studies were found irrelevant, leaving 113 studies available for full-text review. Full texts of 113 papers were retrieved and reviewed, 86 papers were excluded. The search was updated in December 2024 and no further relevant studies were identified. The result was a final inclusion of 27 papers in this review. The reference list of these 27 papers was also screened for any further relevant papers; however, no additional papers were found (please see Fig 1:Study selection Prisma diagram).

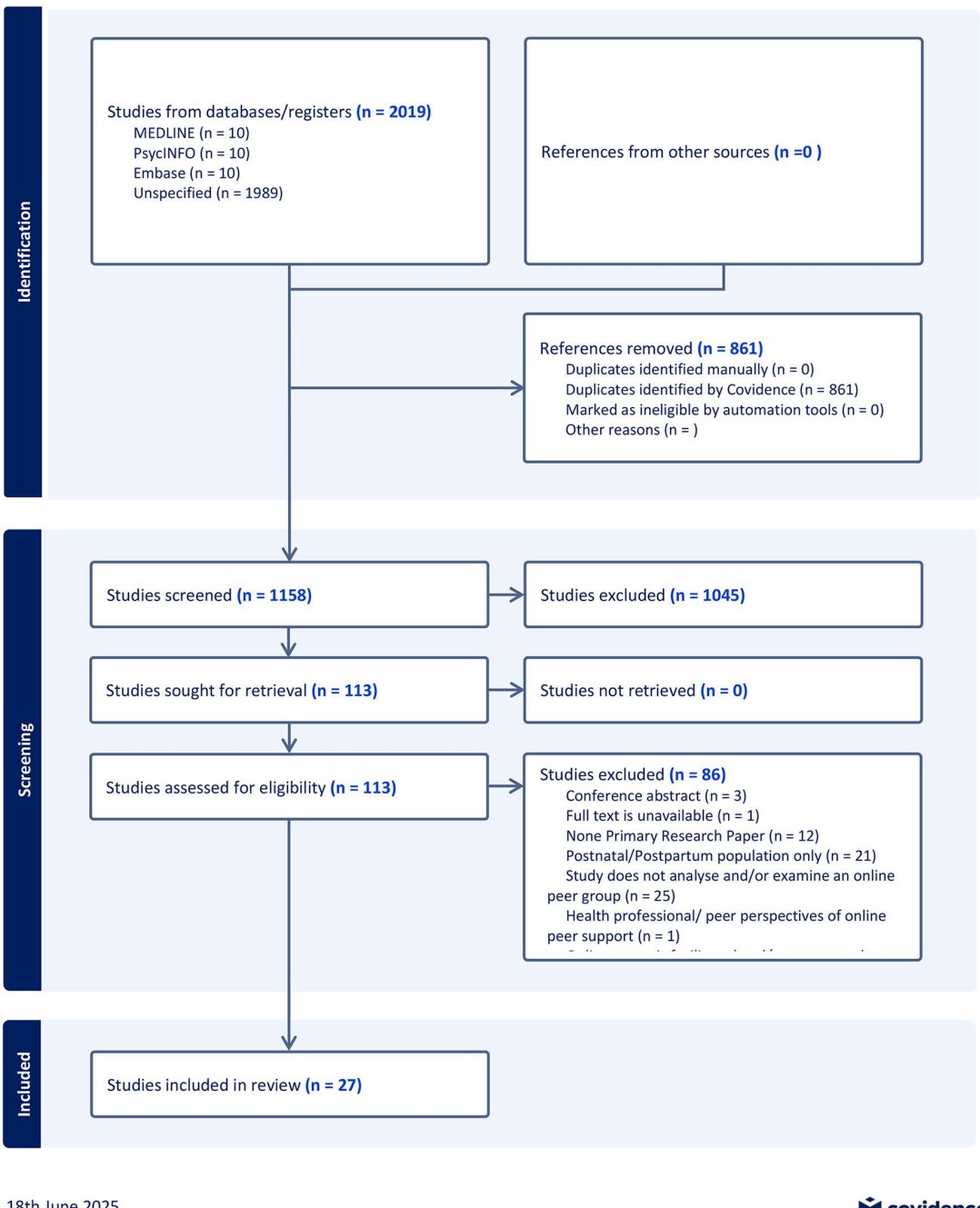

**Fig 1. Prisma diagram.**

## Overview of included studies

Table 2 summarises the key features of the included studies. Studies were published between the years 2002–2024 and were conducted in the United States of America 14,15, 18,20, 26,29,34, 39,40], South America [21], Asia [22–24,30,41,42], Africa [16,17,25,28], Europe [27,31,33,38], the United Kingdom [19], and Australia [13]. One study was transnational

**Table 2. Study characteristics table.**

| Author/ Year/ Country Data Sourced | Aims/ Objectives | Methodology/ Outcome Measure | Population |
|---|---|---|---|
| Adler & Zarchin, 2002 USA [15] | Explore the effectiveness of peer support group for pregnant women on home bedrest. | Qualitative- Exploratory, descriptive investigation | n = 7 Prenatal only |
| Amaro et al., 2023 Brazil [21] | To assess the efficacy of a virtual initiative to support pregnant and postpartum women | Quantitative- Survey Exploratory, descriptive investigation | n = 6942 Prenatal and Postnatal |
| Dai et al., 2022 Wuhan, China [24] | To explore Chinese pregnant women's experience in the Internet-based CenteringPregnancy management model. | Qualitative Interpretative phenomenological . | n = 9 Prenatal Only |
| Dean et al., 2012 South Africa [16] | Investigate the feasibility of mobile phone text messaging (SMS) to promote adherence to antiviral therapy in HIV-infected women recently diagnosed with HIV. | Mixed-Methods Feasibility/ Pilot | n = 7 Prenatal only |
| Duffecy et al., 2022 USA [43] | To evaluate and compare the impact of two versions of an online intervention on the prevention of PPD among people with subclinical depressive symptoms during pregnancy. | Quantitative Randomised Control Trial | n = 210 Prenatal Only |
| Fiks et al., 2017 USA [20] | The article describes a randomized controlled trial (RCT) of Grow-2Gether, to examine its feasibility and acceptability on a larger scale and to test its impact on behaviours. | Quantitative -Randomised Control Trial | n = 87 Prenatal and Postnatal |
| Flax et al., 2022 Nigeria [25] | To measure the impact of the intervention on early initiation of breast-feeding and exclusive breastfeeding at 6 and 24 weeks. | Quasi-experimental longi-tudinal cohort study | n = 1200 Prenatal and Postnatal |
| Gleeson et al., 2022 Australia [13] | To understand the experiences of participation and support for mem-bers of a closed online mother's group (OMG) | Qualitative Interpretive design | n = 12 Prenatal and postnatal |
| Holtz et al., 2015 U.S., Canada, Australia, U.K. [32] | To determine the motivations for using a Facebook page focused on pregnancy and motherhood | Quantitative Survey | n = 647 Prenatal and Postnatal |
| Jiang & Zhu, 2022a China [22] | To investigate the effect that joining multiple peer groups has on infor-mation exchange. | Quasi experimental | n = 24,705 Prenatal only |
| Jiang & Zhu, 2022b China [41] | To investigate whether perceived social support from online communi-ties related to maternal mental health. | Quantitative Survey | n = 500 Prenatal and postnatal |
| Kouri et al., 2006 Fin-land [33] | To describe how pregnant families used a discussion forum called Net Clinic and the core issues discussed by the parents in this virtual environment | Qualitative | n = 21 Prenatal only |
| Lei et al., 2022 China [23] | To examine how pregnant women in different roles in the online com-munity conduct online self- disclosure to obtain social support during the COVID- 19. | Quantitative Mixed methods? | n = 814 posts Prenatal only |
| Linden et al., 2018 Sweden [27] | To report on the effectiveness of this web-based support in terms of improved self efficacy of diabetes management and general well-being in women with Type 1 diabetes during pregnancy and up to six months after childbirth | Randomised Control Trial | n = 174 Prenatal and Postnatal |
| Leiferman et al., 2023 USA [26] | Assessed the feasibility of My Baby, My Move+(MBMM+) that aims to increase prenatal physical activity, enhance mood and sleep hygiene, and reduce EGWG. | Feasibility Randomised Control Trial | n = 49 Prenatal Only |
| Mattson & Ohlendorf., 2023 USA [14] | To examine the role of engagement with online communities by women using medication-assisted treatment (MAT) to manage recovery from opioid use disorder (OUD) during pregnancy and the first year after birth | Qualitative- Secondary Data Analysis | n = 10 Prenatal and Postnatal |
| McCarthy et al., 2020 UK [19] | To investigate how the informational aspects of midwife moderated groups impact on the development of relational continuity. | Qualitative . | n = 31 Prenatal only |
| Nguyen., 2023 USA [34] | Sheds light on how such mothers use social media groups for social support seeking/providing regarding health utilization during their acculturation process. | Qualitative | n = 18 Prenatal and Postnatal |

*(Continued)*

**Table 2.** (Continued)

| Author/ Year/ Country Data Sourced | Aims/ Objectives | Methodology/ Outcome Measure | Population |
|---|---|---|---|
| Patel et al., 2018 Kenya [28] | To create a low-cost service for social support with the aim of improving health-seeking behaviours among women who presented for individual antenatal care | Mixed Methods Feasibility Study | n = 50 Prenatal |
| Perera et al., 2023 USA [39] | This study compared perceived social support among women of all body mass index (BMI) categories with an attempt to assess the efficacy of the BumptUp mobile application to improve social support for exercise during pregnancy and postpartum. | Randomised control pilot trial | n = 35 Prenatal and Postnatal |
| Ronen et al., 2024 USA [29] | To evaluate a pilot study of the IMAGINE intervention, assessing the uptake, acceptability, and utility of IMAGINE. | Mixed-Methods- Single arm pilot study | n = 10 Prenatal and Postnatal |
| Salonen et al., 2014 Finland [38] | To evaluate the contribution of an internet-based intervention to mothers' perceptions of parenting satisfaction, infant centrality and depressive symptoms during the first postpartum year | Quasi-experimental and longitudinal design | n = 1300 Prenatal and Postnatal |
| Shorey et al 2023 Singapore [30] | To explore the perspectives of parents during the perinatal period amid the COVID-19 pandemic and explore the experiences of Singaporean parents receiving perinatal support via the Supportive Parenting App (SPA). | Qualitative- Semi-structured one-to-one interviews | n = 33 Prenatal and Postnatal |
| Simpson et al., 2021 Zambia [17] | The aim of this study, Project Insaka, was to assess the feasibility and acceptability of a mobile phone based support group intervention for HIV positive pregnant women aged 15–24 in Lusaka, Zambia. | Mixed-Methods Before-After design . | n = 61 Prenatal only |
| Skelton et al., 2020 United States [18] | To explore the utilization of an existing pro-breast feeding Facebook group within the context of a virtual COP [community of practice] and how utilization may influence breastfeeding- related knowledge, attitudes, and behaviours. | Mixed-Methods Sequential exploratory design | n = 343 Prenatal and Postnatal |
| Van Haeken et al 2024 Netherlands [31] | The aim of this study was to examine the potential effectiveness of the intervention in enhancing resilience and promoting maternal mental health | Quasi-Experimental | n = 102 Prenatal and Postnatal |
| Wu & Hung., 2019 Taiwan [42] | The study aimed to investigate the effect of exclusive peer-to-peer virtual support on pregnant women's well-being including physical symptoms, depression, social support, maternal attachment, and pregnancy adaptation | Quasi-experimental | n = 121 Prenatal only |

across the United States of American, Canada, Australia and the United Kingdom [32].The studies were categorised into the following groups according to the main topic of the study: General health and/or Advice [13,19,22,24,32–34], Disease Management [16,17,27], Breastfeeding (2 studies) [18,25], Weight Management [26,39], Medicinal Drug use (1 study) [14] and Others [15,20]. The main topic of the two studies in the 'Others' category was bedrest management and infant feeding practices. Ten studies were categorised under 'Well-being' [21,23,28–31,38,40–42]; in the context of this scoping review, well-being is a general term to categorise studies focusing on the mental and emotional needs of perinatal women.

Whilst most studies were based in the United States, six studies indirectly referenced the low national income of the country in which their study was set [16,17,20,22,25,28]. However, there was also evidence that lower socio-economic groups had been targeted in high socio-economic countries. Of all the studies included, eight directly referenced the socio-economic status of participants [15,17,18,20,21,24,29,39]. Of these eight, two studies actively targeted low income participants within the USA [20,29]. In twelve studies, socio-economic status was implicit or indirectly referenced and seven studies suggested that the population of the study were from a lower economic status and this included studies based in the US, Europe and the UK [14,16,19,28,31,34,42]. Seven studies did not reference socio-economic status [13,22,23,27,30,32,33].

The overall quality of the studies included in this scoping review varied across study designs, The five randomised controlled trials (RCTs) were of moderate quality, the most common weakness was participant and investigator blinding due to the nature of the interventions. Of the five quasi-experimental studies, four were of moderate quality, while one was rated as low quality due to limited control over participant similarities and follow-up. Out of the eight qualitative studies, six were moderate in quality, with gaps in addressing researcher reflexivity and relationships with participants, while the other two [13,14] demonstrated high rigor. Overall, the mixed-methods studies showed moderate quality, with strengths in integration and data interpretation but some weaknesses in sampling strategies and nonresponse bias. Four studies had cross-sectional quantitative designs, for which the quantitative section of the MMAT was applied; they were of moderate quality. See S4 File for full results of quality appraisal.

Of the 19 interventional studies, an online peer support group was the only intervention used in 11 studies. In eight studies online peer support was combined with other interventions. In these studies, a peer support group was most often accompanied by a learning resource such as an information database or structured curriculum. Active group moderators were only recorded as being present in 14 out of the 19 studies. In these studies, the active moderators were non-study participants such as researchers or health professionals, who participated in the group by initiating group discussions or answering/asking questions.

The most commonly used virtual/online environment in the included studies was the social media platform Facebook – using either private or open groups (9). Five studies engaged with online discussion forums including Babytree.com (one study) and Netclinic.co.za (one) and a further three referenced unspecific 'online communities'. Nine studies used web or smartphone applications: WhatsApp (three), WeChat (one), RocketChat (one), Zoom (one), Slack (one) and Bumpt App (one) and SunnySide (one) which are an internet-based interventions. Other virtual environments included SMS messaging groups (one) and email communication loops (one).

Of the total 27 studies, 12 included a prenatal population only, while 15 included both a prenatal and postnatal population. In interventional studies of prenatal-only populations, the duration of the intervention varied, with the shortest being four weeks and the longest encompassing the entire pregnancy, approximately 40 weeks. In interventional studies that included a prenatal population and a postnatal population, the intervention commenced in the prenatal period, concluding during the postnatal period; the shortest duration was five months and the longest 15 months.

### Findings synthesis

Our review suggested that peer support had positive effects for both prenatal and postnatal people. In particular, the peer support groups offered a supportive community for individuals, helping to alleviate isolation [13,15–18], reduce stigma related to health conditions such as HIV [16], improve infant feeding behaviours [20] and create a space for friendships to develop [13]. Three main themes identified as being beneficial for optimising online peer support for pregnant people were group size, the role of moderators, and the use of privacy within groups.

### Group size is an important factor in group efficacy

Group size ranged from seven to 24,705. Interventional studies reported that groups where there were a smaller number of study participants created an environment for participants to more easily contribute to group discussions. They were also able to provide beneficial feedback to one another which enabled participants to form genuine connections within the group. However, the optimum number of participants for a 'small' group varied between studies. For example, Adler & Zarchin purposely selected seven participants for their 'focus group', an early example of an online peer support group conducted via an email loop [15]. They reported that this enabled each member to participate fully in group discussions. McCarthy et al divided their 31 participants into one group of 14 and another of 17 which they found enabled the members to 'get to know one another' whilst 'maintaining workable interactions' [19]. Like McCarthy et al other studies recruited a larger cohort of participants and then divided them into smaller groups and this was often on the basis of gestational age.

For example, Fiks et al [20] separated their 87 recruited participants into four peer groups of 9–13 people, Amaro et al [21] recruited 6942 participants to their Godmother Project but then created individual groups with a maximum of 30, and Jiang & Zu [22] assigned their 24,705 participants to smaller peer support groups (the size of the groups was not recorded in the study), all based on gestational age. This was to ensure a higher level of relevance for participants during group interactions.

Observational studies showed that within larger groups, there was a greater chance of lurking behaviour and information overload. 'Lurking', where participants were part of a group but did not actively contribute, limited the benefits of peer support. A survey conducted by Skelton et al., with participants recruited from a large open Facebook page (group) with >6300 members reported that participants rarely actively interacted with the group [18]. This was because they could normally find the answer to their question by searching through past comments. Lei et al [23] and Gleeson et al [13] both also noted that in the large discussion forum many users did not participate in discussions. Furthermore, Gleeson et al., highlighted that large online groups can create information overload [13]. This was the impetus for 12 members of a much larger group (approx. 300 members) to break away. The smaller group offered a sustainable load which allowed participants to keep up with the number of posts and efficiently engage in group discussion.

## Moderator activity and use of further information

Fifteen studies utilised moderators [15–21,24–31]. Moderators varied in their expertise and their roles within the group. In Dai et al's [24] study the moderators were drawn from the research team and their function in the groups was to guide group discussions and answer technical medical questions that participants were unable to determine through discussion. Fiks et al used a psychologist as a moderator, although their specific role was not explained by the study; they reported that group members found their presence helpful [20]. Leiferman et al used moderators who were recruited from the local community [26]. Referred to as 'peer leaders', they were a bridge between the study team and the participants, supporting participants during the study intervention and reporting back on their progress.

Moderators were reported as serving many purposes within the individual studies. For example, Leiferman et al noted that the peer moderators were a key source of support for the women involved in the intervention [26]. In Ronen et al [29] and Fiks et al [20], moderators were important for encouraging engagement within groups and would follow up with participants who were not active in the group by sending them private reminders to join group conversations. Amaro et al [20] and McCarthy et al [19] also noted that the midwife moderators that were engaged in both interventions were able to recognise when women were in need of more individualised attention and offer further support.

It was also important that moderators remained active in the groups as this aided engagement and allowed participants to discuss topics in more depth. For example, Simpson et al reported that in groups where moderators were inactive, topics among participants remained superficial while participants in groups with active moderators discussed more intimate issues with one another [17]. Moderators were also able to monitor the quality of the information in the groups, as evident in McCarthy et al's study [19], and also share their own personal experiences to encourage the participants of the group to share theirs. A combination of these functions was especially useful in Dean et al. and Simpson et al. in which the populations investigated were HIV positive pregnant people in African countries where there is a high stigma surrounding the disease [16,17]. The studies found that by having moderators within the group who could identify and relate with the participants, participants felt more comfortable and safer opening up and discussing their own experiences with others in the group. In addition, by monitoring the quality of the information in the group, participants felt more confident that the information shared with one another was accurate and trustworthy, as the moderators would step in to correct inaccurate or misleading information. Finally, the importance of moderators in responding to risk issues within the group is emphasised by Dean et al as a participant was withdrawn from the group after an urgent referral by the moderators for treatment for severe depression [16].

## Informational privacy allows participants to communicate more comfortably

Informational privacy relates to an individual's right to decide how and to what extent information about themselves is provided to other persons. The importance of privacy in online peer groups was highlighted by Holtz et al [32], who described that the use of an open Facebook group acted as a perceived barrier for 20% of participants. Participants stated that sometimes they would want to respond to a question or post a comment, but as the group was open, those in their own Facebook network could see what they posted, so they felt hesitant to directly interact with the group, especially if the topic was of a personal or sensitive nature. Similarly, whilst Van Haeken et al [31] did not confirm if users were encouraged to remain anonymous, they found that participants were more open to sharing their experiences in the increased anonymity of an online group. Anonymity was particularly important in online support groups addressing stigmatised issues. Dean et al found that privacy facilitated the success of an SMS messaging support group for pregnant people with HIV noting that the women involved would not have engaged with the group had it been in person owing to the stigma around the condition [16].

## Types of online peer support

The following section describes how the results (from the 15 studies that reported on this) were mapped to types of social support as set out in the Conceptual Analysis of Social Support [11]. See Table 3.

## Emotional support

Emotional support was the most common form of social support demonstrated by participants in online peer groups. Many of the participants found their online group provided them with an environment where they were part of a network of similar individuals, they were able to discuss topics openly and they felt the other members understood them well. For example, Dean et al described that participants felt cared for and that members of their group could empathise with their situation [16]. Similarly, Ronen et al found that the participants in the IMAGINE study felt heard and supported in the online peer support group [29].

As participants felt cared for in the groups, they were able to use the space to vent their negative feelings and emotions. This was reflected by Kouri et al [33] as participants would often use the group as an outlet for their concerns or low

Table 3. Online peer support mapped to types of social support.

| | Emotional Support | Informational Support | Appraisal Support | Instrumental support |
|---|---|---|---|---|
| Adler and Zarchin, 2002 | ✓ | | ✓ | |
| Dai et al, 2022 | ✓ | ✓ | | |
| Dean et al, 2012 | ✓ | ✓ | | |
| Denton et al, 2020 | ✓ | ✓ | ✓ | |
| Gleeson et al, 2022 | ✓ | ✓ | | |
| Kouri et al, 2006 | ✓ | ✓ | ✓ | ✓ |
| Leiferman et al., 2023 | ✓ | ✓ | ✓ | ✓ |
| McCarthy et al, 2020 | ✓ | ✓ | | |
| Mattson and Ohlendorf, 2023 | ✓ | ✓ | ✓ | |
| Nguyen, 2023 | ✓ | ✓ | | |
| Patel et al., 2018 | ✓ | ✓ | | |
| Ronen et al., 2024 | ✓ | ✓ | | |
| Skelton et al, 2020 | ✓ | ✓ | ✓ | |
| Simpson et al, 2021 | ✓ | ✓ | ✓ | |

moods. The online group provided the members with a sense of being valued for example [16,17] described that participants would keep an intimate memory of what their peers were going through, and they would follow up with them and ask how they were coping or how things were progressing.

### Informational support

Informational support was the second most common form of social support provided by participants in the online peer groups. Patel et al found that informational support was the most evident benefit of the group as women valued the information sharing between group members [28]. Participants often sought informational support in their peer groups during a waiting period between medical appointments, or if they felt their question was not of medical urgency. Nguyen found that women sought information from the Facebook group '*while waiting for the situation to change or for appointments to be scheduled*' [34, p7]. Dean et al [16] and Simpson et al [17] described that participants would often use the group to discuss their symptoms as this allowed them to check if they should follow up with their doctor. Learning from one another from personal experience rather than from information online was also appreciated. For example in Dai et al participants gained information by discussing questions around pregnancy with each other [24].

### Appraisal support

Appraisal support was demonstrated in nine of the included studies. Participants in these studies were often affirmed by their peers, for example, participants in Nguyen [34] would respond to posts, letting the original poster know that they have gone through similar situations and reassuring the original poster that they were taking the right course of action. Adler & Zarchin [15] and Skelton et al [18] described that participants felt empowered to not give up because of the encouragement they received from their online peers. Similarly for the individuals interviewed by Mattson and Ohlendorf [14], hearing successful stories of recovery management gave the participants encouragement to continue on their recovery journey. Interacting with others in similar situations enabled them to more confidently advocate for themselves and their babies.

### Instrumental support

Instrumental support in the form of provision of tangible goods and services was one of the least common forms of support demonstrated by participants in online peer groups. Participants in Kouri et al provided instrumental support by offering assistance such as sharing car rides [33]. In Leiferman's study, participants were provided with a yoga mat and a scale to aid them in their engagement with the programme. [26]

## Discussion

This scoping review identified 27 papers relevant to the topic of online peer support for perinatal and postnatal women. The findings suggest that online peer support groups offer significant benefits, including emotional, informational, and appraisal support, with structural factors such as group size, moderation, and privacy playing a central role in determining how they are perceived and valued. To contextualise these findings, we compare them with the broader literature on peer support in pregnancy and online peer support groups across different populations, considering potential mechanisms of impact and how they align with the structural factors identified in this review.

### Organisation and structure of groups

Key structural factors, including group size, privacy settings, and active moderation, emerged as critical to the success of online peer support groups. Included studies such as Fiks et al and McCarthy et al) divided participants into small groups to foster more intimate discussions and emotional connections [19,20] This is consistent with broader literature such as

Yamashita et al's systematic review of parenting communities highlighted that small, focused groups in parenting communities enhanced participants' sense of connection and reduced the likelihood of information overload [35].

Privacy settings played a pivotal role in creating safe environments for discussing sensitive topics. Holtz et al. observed that open groups hindered participation due to fears of exposure, whereas closed or anonymized groups encouraged sharing and trust [32]. This aligns with Lin and Shorey's findings in infertility support communities, where participants valued anonymity as a mechanism for fostering honest and open dialogue, particularly in the context of stigmatized conditions [36].

Active moderation emerged as another important factor. Dai et al [24] and Simpson et al [17] highlighted the importance of moderators in ensuring the accuracy of information and fostering respectful discussions. Lin and Shorey similarly noted that moderators mitigated the risks of misinformation and collective negative emotions in infertility groups, suggesting a universal need for skilled facilitators across peer support settings [36]. To maximise the benefits of online peer communities, moderators need training not only in fostering group dynamics and addressing misinformation but also in the unique skills required for online facilitation. This includes proficiency in managing digital platforms, addressing privacy concerns, and navigating asynchronous communication. Providing moderators with both interpersonal and technical training can ensure that they are well-equipped to manage these communities effectively and foster meaningful engagement.

## Types of support and mechanisms of impact

Emotional support was the most frequently reported benefit. Informational support, bridging gaps in formal healthcare, was also a key benefit. This aligns, a systematic review of online health communities, which emphasised how online forums enabled practical, tailored advice to meet users' specific needs [37]. Similarly, Yamashita et al. reported that such platforms fostered a sense of belonging and reduced isolation while prioritizing emotional and informational exchanges [35].

Appraisal support, characterized by validation and reassurance, was highlighted in studies such as McCarthy et al [19]. Participants reported feeling more confident and empowered after receiving peer validation. Instrumental support was less commonly reported, reflecting the inherent limitations of virtual platforms in providing tangible resources. Yamashita et al similarly found that online parenting groups prioritized emotional and informational exchanges over material assistance, suggesting that digital platforms are more effective for psychological support than practical aid [35].

The mechanisms underlying the benefits of online peer support are multifaceted, involving individual and collective dynamics. Due to this variance in the structure and composition of groups, success can be more uniformly considered through reference to the Social Support Scale. Online peer support gave participants a source of emotional support that was not available to them in their day to day lives. Bringing together people with similar experiences and creating a safe space for them to interact, communicate and seek advice, participants were able to offer each other emotional support which was also reinforced by appraisal support as the participants benefitted from the positive reinforcement they received from the group. The emotional support element of the online peer support groups was reinforced in some cases by the presence of moderators and moderators were also a key source of informational support. However, informational support between group members was also highlighted as a beneficial aspect of online peer support. Informational support bridged gaps in healthcare systems, fostering informed decision-making, as described by Dai et al [24]. Participants were able to offer each other anecdotal information in the periods between medical appointments and acted as a sounding board for concerns, rather than having to rely on and sift through the large volume of information available online. The strength of the groups was not found in their ability to offer material support to the women, which was in some cases another aspect of the studies' interventions. Instead, online peer groups provided a space for women to promote their own self efficacy and find comfort and support through mutual contributions to the group, based on shared experiences However, in their systematic review Lin and Shorey, highlight the dual nature of these mechanisms, noting that while online peer support fosters community and mutual benefit, it may also perpetuate misinformation or collective distress in poorly managed

settings [36]. The incorporation of skilful and digitally-competent moderators can play a pivotal to ensure online peer support remains supportive, reliable and effective

## Strengths and limitations

This scoping review was strengthened by the incorporation of the Langford Conceptual Analysis of Social Support [11] to systematically assess the type of support that online peer support might facilitate, thus providing insight into potential mechanisms of effect. It also has several limitations. First, reliance on self-reported data introduces potential biases, as participants' responses may reflect subjective perceptions rather than objective outcomes. Second, the heterogeneity of study designs and outcomes limits the generalisability of findings preventing a more quantitative synthesis.

Included studies primarily focus on reporting the nature of support provided, with less emphasis on the mechanisms of impact or organizational aspects such as group size and the training of peer moderators. Future research should include well-designed studies with longer follow-up periods to better understand the sustained effects of these interventions and the factors influencing their success.

In studies where peer support was combined with other interventions, the studies did not specifically analyse the efficacy of each element of the intervention individually. This means that it is not possible to assess the success of the peer support in isolation. In addition, many of the studies were rated as being of moderate quality, therefore the findings should be interpreted with caution.

## Conclusions

This scoping review provides a comprehensive overview of the potential benefits and structural factors influencing the success of online peer support for pregnant people. Emotional, informational, and appraisal support emerged as central themes, with structural elements like group size, moderation, and privacy playing pivotal roles in fostering engagement. While existing studies emphasise the nature of support provided, future research should prioritise exploring further the mechanisms of impact and evaluating organisational aspects, such as the skills and training of moderators. In addition there is scope to explore the utility of this kind of support in other aspects of pregnancy care, such as smoking cessation.

## Supporting information

**S1 File. PRISMA-ScR-fillable-checklist.**
(PDF)

**S2 File. Search strategy.**
(DOCX)

**S3 File. Data extraction items.**
(DOCX)

**S4 File. Quality appraisal tables.**
(DOCX)

## Author contributions

**Conceptualization:** Maria Duaso, Marilyn Ijeomah-Orji, Lisa Akester, Jenny Roddy, Jayne Samples, Kate Heighway, Nigel Simpson, Tomasina Stacey.

**Data curation:** Honor Morris, Maria Duaso, Marilyn Ijeomah-Orji.

**Formal analysis:** Honor Morris, Maria Duaso, Marilyn Ijeomah-Orji, Tomasina Stacey.

**Funding acquisition:** Tomasina Stacey.

**Investigation:** Maria Duaso, Tomasina Stacey.

**Methodology:** Marilyn Ijeomah-Orji, Tomasina Stacey.

**Project administration:** Tomasina Stacey.

**Supervision:** Maria Duaso, Tomasina Stacey.

**Validation:** Maria Duaso.

**Writing – original draft:** Honor Morris.

**Writing – review & editing:** Maria Duaso, Marilyn Ijeomah-Orji, Lisa Akester, Jenny Roddy, Jayne Samples, Kate Heighway, Nigel Simpson, Tomasina Stacey.

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
