## [Decision Letter · Decision Letter 0]

12 Jun 2025

PLOS ONE

Dear Dr. Stacey,

Thank you for submitting your manuscript to PLOS ONE. After careful consideration, we feel that it has merit but does not fully meet PLOS ONE’s publication criteria as it currently stands. Therefore, we invite you to submit a revised version of the manuscript that addresses the points raised during the review process.

We look forward to receiving your revised manuscript.

Kind regards,

Rana Islamiah Zahroh, PhD

Academic Editor

PLOS ONE

Journal Requirements:

2. We note that your Data Availability Statement is currently as follows: All relevant data are within the manuscript and in Supporting Information files.

4. Please ensure that you include a title page within your main document. You should list all authors and all affiliations as per our author instructions and clearly indicate the corresponding author.

5. Please ensure that you refer to Figure 1 in your text as, if accepted, production will need this reference to link the reader to the figure.

6. Please include a caption for figure 1.

7. Please include a copy of Table 2 which you refer to in your text on page 7.

Additional Editor Comments:

Editor's comments:

1) Please include the period of the literature search in the abstract. Additionally, clarify the inclusion criteria with respect to study types (e.g., quantitative, qualitative, mixed methods) within the abstract’s methods section.

2) Please clarify or provide concrete examples of what is meant by "emotional support" to ensure consistent understanding among readers.

3) While the review aims to explore the types, contexts, and effectiveness of online peer support, the findings primarily focus on factors influencing good conduct. Please provide a more comprehensive synthesis that also addresses the types of peer support and their effectiveness, or clarify the scope of the findings accordingly.

4) Add clear subheadings for each methodological component, including: Study Selection, Data Extraction, Critical Appraisal Assessment (currently embedded and unclear, see around line 98).

5) Clearly report if any software tools (e.g., NVivo, Excel) were used during data extraction or analysis.

6) The manuscript reports the use of multiple appraisal tools (e.g., MMAT and others). Since MMAT covers RCTs, quasi-experimental, and qualitative studies, please justify why additional tools were needed despite MMAT’s broad applicability. Please also add critical appraisal results as supplementary files

7) Indicate the number of included studies by methodology (quantitative, qualitative, mixed methods) both in narratively in the results section and in the characteristics table.

8) Report the geographic distribution of studies in results section (e.g., number from high-income vs. low- and middle-income countries).

9) Reference all tables and figures appropriately within the text (e.g., Figure 1 is uploaded but not cited).

10) Use descriptive subheadings in the results section to guide readers especially before the findings synthesis section (e.g., “Overview of Included Studies,” “Types of Online Peer Support,” etc.).

11) Ensure findings are consistently supported by appropriate references from the included studies. For example, in lines 187-192, citations should be added to show which included studies support the stated themes or observations.

Please also address the reviewers' comments below.

Reviewers' comments:

Reviewer's Responses to Questions

**Comments to the Author**

1. Is the manuscript technically sound, and do the data support the conclusions?

Reviewer #1: Partly

Reviewer #2: Partly

2. Has the statistical analysis been performed appropriately and rigorously?

Reviewer #1: I Don't Know

Reviewer #2: Yes

3. Have the authors made all data underlying the findings in their manuscript fully available?

Reviewer #1: Yes

Reviewer #2: Yes

4. Is the manuscript presented in an intelligible fashion and written in standard English?

Reviewer #1: Yes

Reviewer #2: Yes

Reviewer #1: Congratulations to the authors for their work. This manuscript presents a critical study, but, there are some points that would benefit from further elaboration to enhance the paper's clarity.

1) Methods

a. Please provide a brief explanation of the terms used in the search strategy and include the full search strategy as an appendix.

b. It appears that the authors have included studies with various study designs. Please clarify this in the Methods section.

c. Are there any limitations regarding the publication language or publication year? If so, please specify in the Methods section.

d. How many individuals were involved in the methodological assessment process? Please provide this information in the Methods section.

e. Was any specific software used in synthesizing the data? Additionally, how many individuals were involved in this process?

2) Regarding methodological assessment, most of the included studies were rated as having moderate quality. Could you provide guidance on how readers should interpret the results of this review in terms of the quality of the studies? Please report this in strengths and limitations.

3) Results:

a. The description of the study selection process does not align with the PRISMA diagram. Please address this discrepancy

b. Could you provide further details on the country context of the online peer support? Specifically, was it conducted in high-income countries? If so, please note this as a limitation, as the findings may be more applicable to high-income countries.

c. Please ensure that studies are appropriately cited when reporting their characteristics.

d. In eight studies, online peer support was combined with other interventions. How can we be certain that the observed effectiveness was specifically due to the online peer support and not other interventions?

4) What evidence supports the positive effects of peer support? Did the authors also include results from observational studies to make this determination? Additionally, what criteria were used to assess the effectiveness of peer support?

5) Could you clarify what other types of support are referred to in Table 3?

Reviewer #2: Dear Author,

Your research is highly valuable, and I have a few suggestions that could enhance it.

In the abstract, please state the method more precisely. Clearly indicate the inclusion criteria for the study and mention the databases used.

In the introduction, it is important to clearly articulate the necessity of conducting this research in the final paragraph.

The discussion section could be strengthened. Please make the necessary revisions.

**Do you want your identity to be public for this peer review?** For information about this choice, including consent withdrawal, please see our Privacy Policy

Reviewer #1: No

Reviewer #2: No

---

## [Author Response · Author response to Decision Letter 1]

29 Jul 2025

Comments Response- these have also been submitted as a separate file.

Editor comments

PLOS requires an ORCID iD for the corresponding author in Editorial Manager on papers submitted after December 6th, 2016. Please ensure that you have an ORCID iD and that it is validated in Editorial Manager. This has now been linked/added

Please ensure that you include a title page within your main document. You should list all authors and all affiliations as per our author instructions and clearly indicate the corresponding author. This has now been added

Please ensure that you refer to Figure 1 in your text as, if accepted, production will need this reference to link the reader to the figure. Reference to figure 1 added, p8

Please include a caption for figure 1. Caption added

Please include a copy of Table 2 which you refer to in your text on page 7 Included

Please include captions for your Supporting Information files at the end of your manuscript, and update any in-text citations to match accordingly. The supporting files now include captions and intext citations updated

Please include the period of the literature search in the abstract. Additionally, clarify the inclusion criteria with respect to study types (e.g., quantitative, qualitative, mixed methods) within the abstract’s methods section. This has now been added

Please clarify or provide concrete examples of what is meant by "emotional support" to ensure consistent understanding among readers. This has been added to abstract and methods sections (p7, l134)

While the review aims to explore the types, contexts, and effectiveness of online peer support, the findings primarily focus on factors influencing good conduct. Please provide a more comprehensive synthesis that also addresses the types of peer support and their effectiveness, or clarify the scope of the findings accordingly.

Our original aim included the term effectiveness but during the synthesis we found that effectiveness was rarely directly measured and could not be clearly linked to peer support alone. Many interventions included other components, making it hard to isolate the role of peer support.

Our synthesis has focused on mapping the types of support exchanged, the settings in which they occurred, and the structural factors that shaped group function. We have now clarified this more explicitly in the text by replacing references to effectiveness with perceive benefits to ensure alignment with the aims of a scoping review

Add clear subheadings for each methodological component, including: Study Selection, Data Extraction, Critical Appraisal Assessment (currently embedded and unclear, see around line 98). Subheadings have been added

Clearly report if any software tools (e.g., NVivo, Excel) were used during data extraction or analysis. Covidence data extraction tool was used, this is now included in methodology section

The manuscript reports the use of multiple appraisal tools (e.g., MMAT and others). Since MMAT covers RCTs, quasi-experimental, and qualitative studies, please justify why additional tools were needed despite MMAT’s broad applicability. Please also add critical appraisal results as supplementary files While the MMAT was used for its flexibility in appraising diverse study designs, we also applied design-specific tools to increase appraisal depth and rigour. The CASP checklists (for qualitative and RCT studies) and the JBI checklist (for quasi-experimental studies) allowed us to assess features not covered by the MMAT, such as researcher reflexivity and outcome measurement timing. Full critical appraisal results are provided in the supplementary files.

Indicate the number of included studies by methodology (quantitative, qualitative, mixed methods) both in narratively in the results section and in the characteristics table. This is noted in the narrative on page 9 second paragraph and in Table 2

Report the geographic distribution of studies in results section (e.g., number from high-income vs. low- and middle-income countries). This has been added, page 8

Reference all tables and figures appropriately within the text (e.g., Figure 1 is uploaded but not cited). This has been done, thank you

Use descriptive subheadings in the results section to guide readers especially before the findings synthesis section (e.g., “Overview of Included Studies,” “Types of Online Peer Support,” etc.). This has been added

Ensure findings are consistently supported by appropriate references from the included studies. For example, in lines 187-192, citations should be added to show which included studies support the stated themes or observations. These have been added, thank you

Reviewer #1 comments

Please provide a brief explanation of the terms used in the search strategy and include the full search strategy as an appendix.

There is a reference to the Prospero protocol which has the full search strategy.

Further explanation has been added to line 107-8. There is also a full search strategy now attached as an appendix

It appears that the authors have included studies with various study designs. Please clarify this in the Methods section.

This has been added to the inclusion/exclusion criteria

Are there any limitations regarding the publication language or publication year? If so, please specify in the Methods section.

Added into Methods p5

How many individuals were involved in the methodological assessment process? Please provide this information in the Methods section Added in to methods p5

Was any specific software used in synthesizing the data? Additionally, how many individuals were involved in this process?

No specific software was used.

This was undertaken by two individuals. This has been clarified in the test

Regarding methodological assessment, most of the included studies were rated as having moderate quality. Could you provide guidance on how readers should interpret the results of this review in terms of the quality of the studies? Please report this in strengths and limitations.

We have now addressed this in the Strengths and Limitations section p19/20. We have suggested to interpret findings with caution, particularly regarding effectiveness.

Results:

The description of the study selection process does not align with the PRISMA diagram. Please address this discrepancy

Thank you. This has been corrected and now aligns

Could you provide further details on the country context of the online peer support? Specifically, was it conducted in high-income countries? If so, please note this as a limitation, as the findings may be more applicable to high-income countries.

The income of the country was only ever indirectly referenced in the articles. However, lines 196-205 have been added to the article to outline that the findings are relevant to lower socioeconomic groups.

Please ensure that studies are appropriately cited when reporting their characteristics.

This has now been actioned.

Further explanation has been added to line 107-8. There is also a full search strategy now attached as an appendix

In eight studies, online peer support was combined with other interventions. How can we be certain that the observed effectiveness was specifically due to the online peer support and not other interventions?

Thank you this insightful comment, as noted above it is true it is hard to distinguish what might have had the effect. That is in part why we did not explicitly assess effectiveness please see above in response to editor comment. We have added a section on this to the ‘Limitations’ (line 469-471)

What evidence supports the positive effects of peer support? Did the authors also include results from observational studies to make this determination? Additionally, what criteria were used to assess the effectiveness of peer support?

As above, during the synthesis we found that effectiveness was rarely directly measured and could not be clearly linked to peer support alone. Many interventions included other components, making it hard to isolate the role of peer support. Our synthesis therefore focused on mapping the types of support exchanged, the settings in which they occurred, and the structural factors that shaped group function.

Could you clarify what other types of support are referred to in Table 3?

Types of support that may not strictly fit into the Langford et al Social Support framework. There were no studies that fell into this category and so it has been removed.

Reviewer #2

In the abstract, please state the method more precisely. Clearly indicate the inclusion criteria for the study and mention the databases used. Thank you for this comment, we would like to add this but were unable to, due to the tight word count restriction for the abstract.

In the introduction, it is important to clearly articulate the necessity of conducting this research in the final paragraph. We hope this is now clearly articulated

The discussion section could be strengthened. Please make the necessary revisions. We are not clear what specifically was required as a revision in this comment, but believe that the discussion section adequately situates the findings in the wider literature/explores potential mechanisms of effect and provides a measured interpretation of the results.

---

## [Decision Letter · Decision Letter 1]

21 Oct 2025

Dear Dr. Stacey,

Thank you for submitting your manuscript to PLOS ONE. After careful consideration, we feel that it has merit but does not fully meet PLOS ONE’s publication criteria as it currently stands. Therefore, we invite you to submit a revised version of the manuscript that addresses the points raised during the review process. Please see the specific points from reviewers at the end of this message.

We look forward to receiving your revised manuscript.

Kind regards,

Rana Islamiah Zahroh, PhD

Academic Editor

PLOS ONE

Journal Requirements:

Reviewers' comments:

Reviewer's Responses to Questions

**Comments to the Author**

Reviewer #2: (No Response)

Reviewer #3: All comments have been addressed

Reviewer #4: All comments have been addressed

2. Is the manuscript technically sound, and do the data support the conclusions?

Reviewer #2: (No Response)

Reviewer #3: Yes

Reviewer #4: Yes

3. Has the statistical analysis been performed appropriately and rigorously?

Reviewer #2: (No Response)

Reviewer #3: N/A

Reviewer #4: Yes

4. Have the authors made all data underlying the findings in their manuscript fully available?

Reviewer #2: (No Response)

Reviewer #3: Yes

Reviewer #4: Yes

5. Is the manuscript presented in an intelligible fashion and written in standard English?

Reviewer #2: (No Response)

Reviewer #3: (No Response)

Reviewer #4: Yes

Reviewer #2: Dear Author,

Your research is valuable, but I have some suggestions that could help improve it:

1. The necessity of the research is not clearly articulated in the abstract.

2. The methodology outlined in the abstract is incomplete. Please specify the databases used and include the inclusion and exclusion criteria.

3. The search strategy in the methodology should be detailed, including the keywords that were searched in the electronic databases.

4. In Table 2, which presents the characteristics of the studies, please ensure that each study is referenced.

5. Consider adding a table to evaluate the quality of the studies.

6. Strengthen the discussion section by providing an analysis of the studies.

7. Lastly, please include the source of funding and any potential conflicts of interest at the end of your manuscript.

Reviewer #3: Thank you for asking me to review this paper, I can see that the authors have acted on the comments made previously.

I have spotted a few typos which need correcting before it is ready for publication:

Line 85: check spelling of 'excellence'.

Line 111: There is an extra 'e' and the fullstop is missing at the end.

Lines 172 & 275: Avoid starting sentences with a number.

Reviewer #4: It was a privilege to review your manuscript entitled, “The role of online peer support in pregnancy: systematic scoping review.” I commend the authors for their effort in conducting this work. The authors put in a lot of thoughtful and thorough consideration into the work. The topic is timely and has the potential to be impactful. The manuscript for review began with responses to previous review recommendations, which were well addressed and incorporated into the body of the work.

I appreciate the clarity of the research question and the contribution it makes to the existing body of knowledge. I also appreciate the opportunity to provide some constructive feedback that may help strengthen the manuscript and improve its clarity and methodology for the global readership.

Overall assessment: This manuscript offers a timely and well-structured scoping review of online peer support in pregnancy, synthesizing 27 studies across diverse geographies and platforms. The mapping of findings to the Langford Conceptual Analysis of Social Support is a particular strength, helping to translate heterogeneous evidence into clear domains (emotional, informational, appraisal, instrumental). The review adds concrete design levers—group size, active moderation, and privacy settings—that have immediate relevance for program developers, clinicians, and digital health implementers. This work will make a valuable contribution to the maternal digital-health literature.

General comments:

The authors alternate between “pregnant people’ and “pregnant women”. Clarify in the Introduction section if there is an intentional purpose for this. Otherwise, choose one term and apply it consistently throughout the manuscript.

Line-by-line comments

Line 85 – Fix the typo “Ecxellence” to “Excellence”.

Line 111 – The reference statement pointing to the supplemental document S2 ends abruptly. The authors should close out the sentence.

Line 129 – In addition to listing the data items considered in this scoping review, the authors should consider adding a Data Items Table and definitions (in supplementary materials) for reference.

Line 188 – The reference against “Weight Management” is missing an opening bracket ( [ ). Ensure bracketed references are correct and consistent.

Line 263 – Consider using a dash (-) to proffer clarity for “9 13 people” as “9-13 people” if the intention is to portray a number range.

Recommendation: Minor revision — The evidence synthesis is solid; improvements mainly concern formatting and correction of spelling errors.

**Do you want your identity to be public for this peer review?** For information about this choice, including consent withdrawal, please see our Privacy Policy

Reviewer #2: No

Reviewer #3: **Yes: ** Julie Abayomi

Reviewer #4: **Yes: ** Babajide Adewumi

---

## [Author Response · Author response to Decision Letter 2]

12 Nov 2025

Response to Reviewers November 2025

Reviewer #2:

1. The necessity of the research is not clearly articulated in the abstract.

Response: We are sorry that the reviewer does not feel that the necessity for the research (review) is clear in the abstract, we have tried to make it as explicit as possible within the limited word count.

2. The methodology outlined in the abstract is incomplete. Please specify the databases used and include the inclusion and exclusion criteria.

Response: Thank you for this comment, we have added the databases, but been unable to include all criteria due to the word count restriction of the abstract.

3. The search strategy in the methodology should be detailed, including the keywords that were searched in the electronic databases.

Response: The detailed search strategy is in S2, which includes all the keywords

4. In Table 2, which presents the characteristics of the studies, please ensure that each study is referenced.

Response: Added, thank you

5. Consider adding a table to evaluate the quality of the studies.

Response: The quality appraisal stables are in S3

6. Strengthen the discussion section by providing an analysis of the studies.

Response: Thank you for this comment, we have provided analysis of the quality of the papers and provided a full synthesis, we do not feel that further individual analysis of each study is not appropriate in this type of review.

7. Lastly, please include the source of funding and any potential conflicts of interest at the end of your manuscript.

Response: The funding source is included on page 3 and a separate conflict of interest statement was submitted to the journal.

Reviewer #3:

1. Line 85: check spelling of 'excellence'.

Response: Corrected

2. Line 111: There is an extra 'e' and the fullstop is missing at the end.

Response: Corrected

3. Lines 172 & 275: Avoid starting sentences with a number.

Response: Corrected

Reviewer #4:

Overall assessment: This manuscript offers a timely and well-structured scoping review of online peer support in pregnancy, synthesizing 27 studies across diverse geographies and platforms. The mapping of findings to the Langford Conceptual Analysis of Social Support is a particular strength, helping to translate heterogeneous evidence into clear domains (emotional, informational, appraisal, instrumental). The review adds concrete design levers—group size, active moderation, and privacy settings—that have immediate relevance for program developers, clinicians, and digital health implementers. This work will make a valuable contribution to the maternal digital-health literature.

Thank you

General comments:

1. The authors alternate between “pregnant people’ and “pregnant women”. Clarify in the Introduction section if there is an intentional purpose for this. Otherwise, choose one term and apply it consistently throughout the manuscript.

Response: Thank you, this has been made consistent through the manuscript.

2. Line 85 – Fix the typo “Ecxellence” to “Excellence”.

Response: Corrected

3. Line 111 – The reference statement pointing to the supplemental document S2 ends abruptly. The authors should close out the sentence.

Response: Corrected

Line 129 – In addition to listing the data items considered in this scoping review, the authors should consider adding a Data Items Table and definitions (in supplementary materials) for reference.

Response: Thank you for this comment, we have included a new supplementary file which include the data extraction tool used (data items).

Line 188 – The reference against “Weight Management” is missing an opening bracket ( [ ). Ensure bracketed references are correct and consistent.

Response: Corrected

Line 257 – Consider using a dash (-) to proffer clarity for “9 13 people” as “9-13 people” if the intention is to portray a number range.

Response: Corrected

---

## [Editor Report · Decision Letter 2]

14 Dec 2025

Title The role of online peer support in pregnancy: systematic scoping review

PONE-D-25-09522R2

Dear Dr. Stacey,

We’re pleased to inform you that your manuscript has been judged scientifically suitable for publication and will be formally accepted for publication once it meets all outstanding technical requirements.

Kind regards,

Rana Islamiah Zahroh, PhD

Academic Editor

PLOS One

---

## [Editor Report · Acceptance letter]

PONE-D-25-09522R2

PLOS One

Dear Dr. Stacey,

I'm pleased to inform you that your manuscript has been deemed suitable for publication in PLOS One. Congratulations! Your manuscript is now being handed over to our production team.

Kind regards,

on behalf of

Dr Rana Islamiah Zahroh

Academic Editor

PLOS One